# Acceptance and Commitment Therapy for Destructive Experiential Avoidance (ACT-DEA): A Feasibility Study

**DOI:** 10.3390/ijerph192416434

**Published:** 2022-12-07

**Authors:** Euihyeon Na, KangUk Lee, Bong-Hee Jeon, Cheolrae Jo, Uk-Hwan Kwak, Yujin Jeon, Kyojin Yang, Eui Jin Lee, Jin Jeong

**Affiliations:** 1Department of Neuropsychiatry, Presbyterian Medical Center, Jeonju 54987, Republic of Korea; 2Department of Psychiatry, Kangwon National University Hospital, Chuncheon 24289, Republic of Korea; 3Mind with Mind Psychiatric Clinic, Changwon 51436, Republic of Korea; 4Maum Gonggam Psychiatric Clinic, Goyang 10497, Republic of Korea; 5Workplace Mental Health Institute, Kangbuk Samsung Hospital, Seoul 03181, Republic of Korea; 6Jeong Jin Psychiatric Clinic, Suwon 16489, Republic of Korea

**Keywords:** acceptance and commitment therapy, destructive experiential avoidance, deliberate self-harm, alcohol use disorder, behavioral addiction, adolescents

## Abstract

Background: This study is a preliminary study on an acceptance and commitment therapy (ACT) program that mitigates destructive experiential avoidance (DEA) behaviors, including self-harm behavior and addiction; Methods: Twenty participants aged 15–25 years who had confirmed DEA behavior within the last month participated in a total of six sessions of ACT. Demographic characteristics, history of psychiatric illness, and TYPES and patterns of DEA behavior were confirmed in the baseline survey. The severity of clinical symptoms, frequency of DEA behavior and impulsivity, characteristics of experiential avoidance (EA) behavior, depression, and quality of life (QOL) were measured before and after the program for comparative statistical tests using the intention-to-treat method. Furthermore, the severity of clinical symptoms was evaluated after each program, along with the frequency of DEA behavior and trends in impulsivity, which were investigated based on the behavior log; Results: After the ACT program, both the frequency of DEA behavior and impulsivity and the severity of clinical symptoms, depression, and anxiety decreased significantly. Furthermore, among the EA characteristics, pain aversion, distraction and inhibition, and delayed behavior significantly improved. Moreover, the overall QOL, psychological and social relationships, and QOL regarding the environment also improved; Conclusions: The results of this feasibility study demonstrate the potential of the ACT program as an effective intervention in DEA behavior. The results of this study may be used as preliminary data for future large-scale randomized studies.

## 1. Introduction

Experiential avoidance (EA) refers to an intentional behavior to avoid or escape from aversive internal experiences such as affect, thoughts, memories, and bodily sensations among private events [1]. In the short term, these intentional efforts may have immediate relief and reinforcing effects; however, in the long term, avoidance results in devastating outcomes of dysfunctioning and decreased quality of life (QOL) [2]. Various studies have reported the adverse consequences of destructive and rigid EA, which are reflected in transdiagnostic mental illnesses such as depressive disorders [3], anxiety disorders [4], obsessive–compulsive disorder [5], substance use disorder, [6] eating disorders [7], and post-traumatic stress disorder [8]. In particular, non-suicidal self-injury may occur in extreme cases of destructive experiential avoidance (DEA) behaviors [9]. In this context, the Research Domain Criteria (RDoC) conducted by the National Institute of Mental Health in the United States defined EA as a behavioral factor in the negative valence system affect and reduced approach motivation and approach behavior in positive valence, which have high treatment approach possibilities [10]. The RDoC also suggests EA as an area to be studied in mental health research and clinical practice [11]. In particular, adolescence is the period when DEA behavior, including self-mutilation behavior, begins, and adolescents have the highest prevalence rates for these behaviors compared with other age groups [12]. Furthermore, adolescents are severely negatively affected on their mental health development and growth due to DEA behaviors [13]. Meanwhile, serious and widespread substance use disorders such as alcohol, tobacco, and opioid analgesics have raised awareness in society in recent years [14]. Moreover, behavioral addictions such as gambling disorder, internet gaming disorder, and eating disorders are also increasing in severity among adolescents [15]. As previous studies indicated, DEA behaviors in adolescents also increase the public health burden, such as rehospitalizations and emergency department visits [16,17]. In this context, it has been required to develop therapeutic interventions for DEA behaviors in adolescents. For instance, a meta-analysis reported the therapeutic possibility of various psychological interventions, including cognitive behavioral therapy (CBT), dialectical behavior therapy (DBT), and mentalization-based therapy (MBT), in reducing non-suicidal self-injury in adolescents [18]. However, there remains an urgent need to identify specific and transdiagnostic approaches to overall DEA behaviors.

Acceptance and commitment therapy (ACT) has been introduced in various mental health areas as one of the third-wave cognitive behavioral therapies [19]. The unique and ultimate goal of ACT is to promote one’s psychological flexibility as a critical process of human adaptation and well-being [20]. Specifically, ACT helps individuals alter the impact of their inner experiences on behavior by using its six core processes: acceptance, cognitive defusion, contact with the present moment, self-as-context, values, and committed action [21]. Through randomized controlled studies, there is empirical evidence of ACT on anxiety disorder and depressive disorder [22]. The efficacy of ACT has also been confirmed in studies on substance use disorder [23], post-traumatic stress disorder [24], and comorbid psychiatric disorders that are considered significant challenges for clinicians and practitioners [25]. Although previous studies have indicated that ACT-based interventions effectively reduce symptoms of each psychiatric disorder, few studies evaluate ACT in DEA as a transdiagnostic factor of mental disorders. Therefore, the development of a new ACT-based treatment program focusing on DEA in various clinical areas and the evaluation of the efficacy of such a program are necessary for advancing the treatment strategies for individuals with DEA behaviors. In addition, cultural differences and various mental healthcare systems in different countries need to be considered [26], and a design based on contextual behavioral science, which is the fundamental philosophy of ACT that fulfills the purpose of the treatment program [27], is required. Through a feasibility study of a structured ACT program that alleviates DEA behaviors for adolescents with clinically significant levels of mental health issues, this study investigated its efficacy and feasibility.

## 2. Materials and Methods

### 2.1. Participants

We conducted a quasi-experimental study, from September 2019 to February 2020, as a feasibility study for a future randomized controlled trial. Participants were in their middle and late adolescence (aged 15–25 years) who had shown DEA behavior for the previous three months and were recruited from several psychiatric clinics in South Korea. The “destructiveness” of experiential avoidance, which could result in functional impairment in the participants, was clinically assessed by fully trained psychiatrists through functional analysis. The major categories for established problematic behaviors of DEA were as follows: (a) Deliberate Self-Harm, (b) Excessive Alcohol and/or Drug use, (c) Binge and/or Restrictive Eating, (d) Sexual Promiscuity, (e) Excessive internet and/or Computer Game Use, (f) Aggression, and (g) Others [28]. Participants with high suicidal risk or acute mental illness who required inpatient treatment were excluded. Other exclusion criteria were a history of brain injury, organic mental disorders, and intellectual disability. Written informed consent was obtained from all participants and their parents or guardians. Of 22 adolescents, 20 (5 males, 15 females) were eligible and consented to participate in this study. This study was approved by the Institutional Review Board (IRB) of Kangwon National University Hospital (IRB number: KNUH-A-2019-08-005).

### 2.2. The ACT-DEA Program

Considering the cognitive developmental characteristics of adolescents, the ACT-DEA program was conducted with six weekly 40-min sessions. Each session was composed of an ACT matrix [29], a tool that applies ACT in structured sessions. Additionally, crisis survival skills training and stabilization techniques that can overcome the urges and cravings of DEA [30,31], such as self-harm and addictive behaviors, were applied in the initial session. Techniques to explore their own values in adolescence were also implemented during the middle sessions [21]. The details of the program were as follows: (1) introduction to the ACT matrix and crisis survival skills training; (2) understanding the functional analysis of behaviors on the ACT matrix and learning mindful grounding techniques; (3) value clarifications and reinforcing stabilization techniques; (4) identifying the effectiveness of DEA behaviors; (5) understanding acceptance using metaphors; and (6) planning commitment behaviors and closing. Each session was conducted by an experienced ACT therapist and psychiatrist, and the progress was managed using a checklist regarding the main topics of each session. A simple assignment was provided based on the experience after each session, and a debriefing session regarding the assignment was held at the beginning of the next session.

### 2.3. Measures

All participants were assessed by two points: baseline (pre-) and post-intervention. The post-intervention assessment was conducted one week after their final program session. In addition, participants completed process measures before the beginning of each session.

The baseline questionnaire aimed to identify their sociodemographic status, clinical histories, and characteristics of DEA(s). The clinical diagnosis was made under *The Diagnostic and Statistical Manual of Mental Disorders, Fifth Edition* (DSM-5) [32]. For a more detailed evaluation of DEAs, we used the Korean version of the Inventory of Statements about Self-Injury (K-ISAS) and the Korean version of the Alcohol Use Disorders Identification Test (AUDIT-K) [33,34]. The K-ISAS was used to assess the lifetime frequency of 12 deliberate self-harm behaviors and their functions [34]. Problematic alcohol use was identified with an AUDIT-K score of 20 for male participants and 10 for female participants [35].

A questionnaire for experiential avoidance and mood symptoms, and a quality of life scale were conducted as outcome measures. The 24-item brief form of the Korean version of the Multidimensional Experiential Avoidance Questionnaire (K-MEAQ-24) assessed the multidimensional aspects of experiential avoidance, including behavioral avoidance, distress aversion, repression/denial, distraction/suppression, procrastination, and distress endurance [36]. The Patient Health Questionnaire (PHQ)-9 and Generalized Anxiety Disorder (GAD)-7 were used to measure depressive and anxiety symptoms [37,38]. The items of each assessment tool were scored on a 4-point Likert scale, with higher total scores representing more severe mood symptoms [37,38]. The Korean version of the World Health Organization Quality of Life Scale Abbreviated Version (WHOQOL-BREF) is a 26-item measure of the quality of life (QOL) with responses given on a 5-point Likert scale [39]. The items assessed subjective responses to overall QOL and four QOL domains including physical health, psychological, social relationships, and the environment [39]

Before starting each session and post-intervention, participants reported their DEA(s) frequency and the urge to engage in the behavior during intersession periods. Simultaneously, therapists evaluated the severity of the symptoms of participants using the Clinical Global Impression Scale (CGI-S) [40]

### 2.4. Data Analysis

All statistical analyses were carried out using SPSS version 24.0 (IBM Corp., Armonk, NY, USA). Given the non-normal distribution of the data shown by the Shapiro–Wilk test, the Wilcoxon signed ranks test was conducted to compare pre-intervention data with post-intervention data. An intent-to-treat analysis was performed using the last-observation-carried-forward method. All data are presented as median (range Q1–Q3) values. The level of statistical significance was set at *p* < 0.05 for all analyses.

## 3. Results

### 3.1. Sociodemographic Data

Of 20 participants, 15 (75.0%) were middle school, high school, and university students (Table 1). More than half (*n* = 11; 55.0%) of the participants were in households of a lower socioeconomic status (SES). None of them had higher SES. Many (*n* = 15; 75.0%) participants met the criteria for depressive disorders in the DSM-5 and 61.9% (*n* = 13) were on psychiatric medication.

### 3.2. Characteristics of Destructive Experiential Avoidance

Most (*n* = 16; 80.0%) participants engaged in two or more DEAs (Figure 1). Nearly all participants (*n* = 18; 90.0%) reported self-mutilation behavior as their DEA (Table 2). It was followed by binge and/or restrictive eating (*n* = 8; 40.0%), aggression (*n* = 6; 30.0%), excessive alcohol and/or drug use (*n* = 4; 20.0%), excessive internet and/or computer game use (*n* = 4; 20.0%), sexual promiscuity (*n* = 3; 15.0%), and others (*n* = 1; 5.0%). 

When identifying self-mutilation behavior from corresponding participants, 72.2% (*n* = 13) responded that their self-mutilation behaviors had an intrapersonal function. In other words, many of them engaged in their DEA to control their private events such as negative ideas or feelings. Meanwhile, all participants with excessive alcohol use met the criteria for alcohol use disorder according to the AUDIT-K.

### 3.3. Characteristics of Destructive Experiential Avoidance

Of the 20 participants, 15 (75.0%) completed the ACT-DEA program, three dropped out due to acute psychiatric hospitalization, and the remaining two for undisclosed reasons. Considerable differences were identified between the pre- and post-assessments of participants regarding their DEA features and clinical characteristics, including K-MEAQ-24, PHQ-9, GAD-7, CGI-S, and WHOQOL-BREF. (Table 3) The post-assessment data indicated a significantly lower frequency and urge to engage in whole DEA(s) and self-mutilation, with lower scores on the total of the K-MEAQ-24, PHQ-9, GAD-7, and GCI-S than the pre-assessment data (all *p* < 0.05). While scores for behavioral avoidance and the repression/denial subdomains of the K-MEAQ-24 were not markedly different between pre- and post-measurements (all *p* > 0.05), the post-measurements of other subdomains including distress aversion, distraction/suppression, procrastination, and distress endurance illustrated more improved scores than those of the pre-measurement data (all *p* < 0.05). In addition, the post-assessment data of most domains of the WHOQOL-BREF had higher scores than those of the pre-assessment data (all *p* < 0.05), except for the score on the physical health domain (*p* = 0.061).

The session-by-session reporting of the participants’ frequencies of the overall DEA(s) and self-mutilation are illustrated in Figure 2. The sum of the behavioral frequencies of corresponding participants showed a decreasing trend as the session progressed.

## 4. Discussion

This study aimed to confirm the validity of the ACT-DEA program for DEA behavior in adolescents with clinically significant symptoms of mental illness. After the ACT-DEA program, the frequencies and urges of overall DEA behavior and self-harm behavior were significantly reduced compared with those before the program. Furthermore, the tendency of behavioral frequency decreased as each session progressed. In addition, clinical features such as depression, anxiety, overall clinical severity, and QOL, as well as most EA characteristics, improved after the program.

Most of the participants in this study were attending school and had lower-middle socioeconomic status in their families. More than half of the participants participated in the program while undergoing medicational treatment. Previous studies have reported that a relatively low socioeconomic status is a factor for mental illness [41], and a community approach is necessary for these environments for the prevention and intervention of adolescent mental illness and mental health risks [42]. Even with medication treatment, many of the subjects who participated in this study continued to engage in multiple destructive behaviors (e.g., self-harm and addictive behaviors, or self-harm and problematic eating behaviors). Considering these characteristics that reflect the severity of the participants, it can be expected that ACT-DEA may be applied to adolescents with challenging clinical situations exhibiting serious self-harming behaviors. Furthermore, most of the participants with relatively severe cases completed the program, which supports the feasibility of ACT-DEA. Moreover, most participants responded that the self-harming behavior function was intrapersonal, which reveals that the ACT approach, which generally addresses intrapersonal processes [43], is effective in DEA, including self-harm and addictive behaviors.

After the ACT-DEA program, overall improvement was observed not only in the K-MEAQ, which measures the characteristics of EA, but also in the PHQ-9, GAD-7, CGI-S, and WHOQOL-BREF, which measure the severity of mental illness and QOL. Among the K-MEAQ subscales, improvements were confirmed in distress aversion, distraction/suppression, procrastination, and distress endurance. This is the result of the effects of the main processes of ACT: acceptance, mindfulness, and committed actions with values [44]. Through ACT-DEA, acceptance with mindfulness is improved, instead of escape/avoidance and the attachment with conceptualized self or past/future; and the tendency to practice behavior consistent with one’s values is increased even in difficult situations [44]. In other words, it was attempted to approach a significant person or object with a willingness to experience psychological pain without using unnecessary control and to practice self-care and self-kindness [45]. Those specific therapeutic processes of the ACT-DEA program may improve the intrapersonal discomforts most participants reported as the reason for engaging in their DEA. However, no significant changes were observed in the areas of behavioral avoidance and repression/denial. One possible cause may be related to the contents of individual items constituting the subscale. For example, behaviors that avoid or reduce risky situations that may trigger DEA have been reported as avoidance in behavioral avoidance [46]. For instance, when an adolescent with addictive behavior attempts to avoid situations (internet cafes, bars, or clubs) or people (relationships with peer groups with addictive behaviors) related to addiction, it is possible that behavioral avoidance is reported. In repression/denial, improvement is likely to be seen when the method of non-judgmental awareness in the present moment is used as in other EA characteristics; however, long-term practice and follow-up are required. These points must be considered when establishing a maintenance treatment plan after the end of the ACT-DEA program, including booster sessions. Furthermore, various aspects of QOL improved after the ACT-DEA program, and it is noteworthy that not only psychological QOL but also social and environmental QOL improved. There are two possible factors that contributed to this: one is through the main techniques of ACT, such as defusion and the self-as-process [44]. In other words, the result of being able to view one’s own environment with an improved balanced viewpoint in contact with the present moment. The second possibility is that one of the commitment behaviors acquired from participating in the ACT-DEA program is the additional improvement effect acquired from increased pro-social behavior and interpersonal interactions in the participants’ daily lives [47]. In this context, the further revision of the ACT-DEA program incorporating interpersonal approaches based on contextual behavioral science, such as functional analytic psychotherapy [48], may contribute to the improvement of the treatment effectiveness [43].

The limitations of this study are as follows: First, as a feasibility study conducted with a small sample size and single-arm, there is a limitation in the generalization of the findings of this study. Second, although most participants had been receiving continuous medication treatment before the program, the treatment plan including medication was not controlled in this study, and it might be possible that therapeutic factors other than the ACT-DEA program itself affected the outcome. Third, the dropout rate of the study might affect the outcome as well. Finally, the long-term effect of the program on DEA and psychological factors could not be definitively established because the interval follow-up assessment was not conducted after the end of treatment.

## 5. Conclusions

Despite these limitations, this feasibility study confirmed the intervention effect of ACT on DEA in adolescent psychiatric patients with various DEA behaviors, including self-harm behavior and addictive behaviors. In particular, this program is expected to be helpful in clinical cohorts with complex behavioral problems. The results of this study may be used as preliminary data to suggest the direction of a large-scale randomized controlled study to validate the efficacy of ACT-based programs in adolescents with complex behavioral problems.

## Figures and Tables

**Figure 1 ijerph-19-16434-f001:**
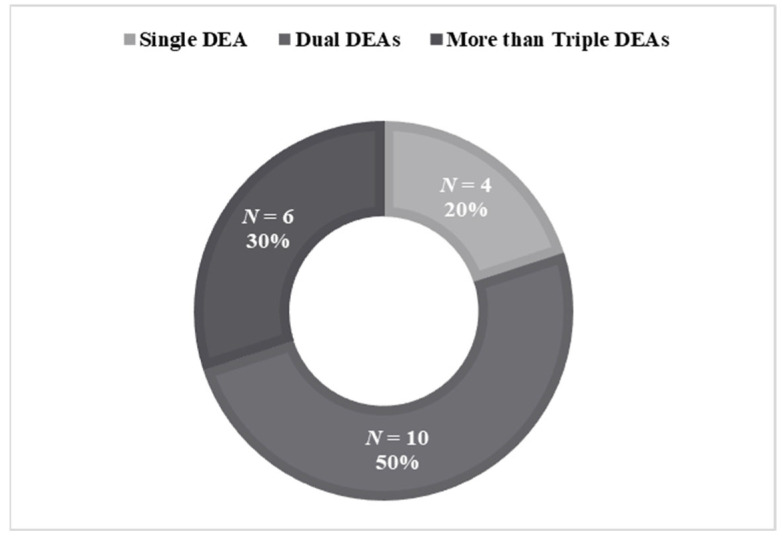
Participants by the number of their destructive experiential avoidance (DEA) behaviors.

**Figure 2 ijerph-19-16434-f002:**
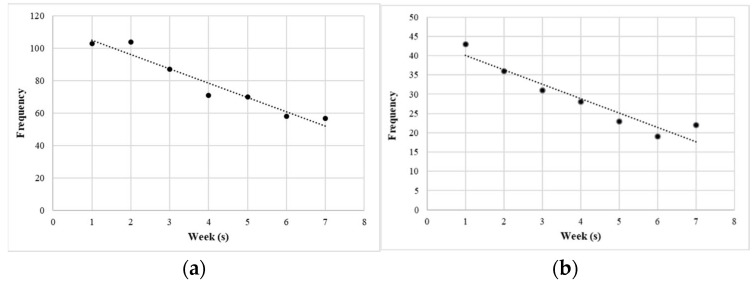
Trends in frequency of destructive experiential avoidance (DEA) behaviors during the study. (**a**) overall DEAs; (**b**) deliberate self-harm.

**Table 1 ijerph-19-16434-t001:** Sociodemographic and clinical characteristics (*N* = 20).

Variables	
Age	18.0 (16.0; 19.8)
Gender (*N*, %)	
Men	5 (25.0%)
Women	15 (75.0%)
Education and employment status (*N*, %)	
Students	15 (75.0%)
Employment	4 (20.0%)
Unemployment	1 (5.0%)
Socioeconomic status (*N*, %)	
Middle	9 (45.0%)
Low	11 (55.0%)
Family history of mental illness (*N*, %)	3 (15.0%)
Past history of mental illness (*N*, %)	11 (55.0%)
DSM-5 diagnostic category (multiple responses; *N*, %)	
Bipolar and related disorders	3 (15.0%)
Depressive disorders	15 (75.0%)
Obsessive–compulsive and related disorders	1 (5.0%)
Trauma and stressor-related disorders	1 (5.0%)
Substance-related and addictive disorders	1 (5.0%)
Feeding and eating-related disorders	2 (10.0%)
Psychiatric medication use (*N*, %)	13 (65.0%)

DSM: *Diagnostic and Statistical Manual of Mental Disorders.*

**Table 2 ijerph-19-16434-t002:** Destructive experiential avoidance (DEA) within the past month (multiple responses).

Variables	
Deliberate self-harm (*N*, %)	18.0 (90.0%)
Duration (Months)	64.5 (9.0; 90.5)
Whole frequency (*N*)	42.5 (16.3; 100.0)
Function of behavior (*N*, %)	
Intrapersonal	13.0 (72.2%)
Interpersonal	3.0 (16.7%)
Both	2.0 (11.1%)
Major form of behavior (*N*, %)	
Skin cutting	9.0 (50.0%)
Excessive scratching	1.0 (5.6%)
Biting	1.0 (5.6%)
Hitting	5.0 (27.8%)
Pinching	1.0 (5.6%)
Eating something harmful	1.0 (5.6%)
Excessive alcohol and/or drug use (*N*, %)	4.0 (20.0%)
Suspected of Alcohol Use Disorder	4.0 (100.0%)
Binge and/or restrictive eating (*N*, %)	8.0 (40.0%)
Sexual promiscuity (*N*, %)	3.0 (15.0%)
Excessive internet and/or computer game use (*N*, %)	4.0 (20.0%)
Aggression (*N*, %)	6.0 (30.0%)
Others (*N*, %)	1.0 (5.0%)

**Table 3 ijerph-19-16434-t003:** Wilcoxon signed rank test results for pre- and post-assessments.

Variables	Pre	Post	Z Wilcoxon	*p*
Destructive Experiential Avoidance				
Whole behavior: frequency	4.50 (1.50; 7.75)	1.69 (0.00; 2.00)	−3.347	0.001 **
Whole behavior: urge	3.17 (2.00; 4.00)	2.07 (1.25; 3.00)	−2.356	0.018 *
Deliberate self-harm: frequency	1.50 (0.00; 4.00)	0.75 (0.00; 1.75)	−2.715	0.007 **
Deliberate self-harm: urge	3.00 (1.25; 4.00)	1.31 (0.00; 2.00)	−2.427	0.015 *
K-MEAQ-24	81.50 (76.50; 89.25)	78.50 (64.50; 87.00)	−2.158	0.031 *
Behavioral avoidance	20.00 (16.00; 21.00)	16.37 (13.75; 18.75)	−1.631	0.130
Distress aversion	18.00 (11.25; 22.75)	14.06 (10.25; 17.00)	−2.515	0.012 *
Repression/denial	12.50 (9.00; 16.00)	14.94 (14.00; 16.00)	−1.496	0.135
Distraction/suppression	15.50 (11.00; 18.75)	11.44 (9.25; 14.00)	−3.585	<0.001 ***
Procrastination	16.00 (15.00; 21.00)	14.81 (13.00; 16.00)	−3.585	0.004 **
Distress endurance	15.00 (11.25; 17.00)	16.38 (14.00; 18.75)	−2.164	0.030 *
PHQ-9	18.50 (12.50; 23.00)	9.50 (5.25; 12.50)	−3.201	0.001 **
GAD-7	13.50 (6.50; 18.50)	6.81 (2.50; 9.75)	−3.204	0.001 **
CGI-S	4.00 (4.00; 5.00)	2.88 (2.00; 3.00)	−3.756	<0.001 ***
WHOQOL-BREF				
Overall	4.00 (3.00; 5.75)	6.00 (5.00; 6.75)	−3.219	0.001 **
Physical health	10.00 (8.57; 10.86)	11.64 (10.43; 13.57)	−1.875	0.061
Psychological health	8.33 (6.67; 10.00)	10.38 (8.33; 11.33)	−2.759	0.006 **
Social relationship	10.67 (8.33; 12.00)	12.50 (12.00; 13.33)	−2.647	0.008 **
Environmental	10.50 (8.25; 12.50)	12.25 (11.50; 13.38)	−2.501	0.012 *

K-MEAQ-24: The brief form of the Korean version of the multidimensional experiential avoidance; PHQ-9: patient health questionnaire-9; GAD-7: generalized anxiety disorder-7; CGI-S: clinical global impressions-severity; WHOQOL-BREF: World Health Organization quality of life scale abbreviated version. * *p* < 0.05; ** *p* < 0.01; *** *p* < 0.001.

## Data Availability

The data that support the findings of this study are available from the corresponding author upon reasonable request.

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
