# Peer review of "Acceptance and Commitment Therapy for Destructive Experiential Avoidance (ACT-DEA): A Feasibility Study"

_ijerph, 2022, doi:10.3390/ijerph192416434_

Round 1

Reviewer 1 Report

The paper reported the intervention effect of ACT on destructive experiential avoidance in youth psychiatric patients. This study is generally well conceptualized and investigates an important issue. 

- When is the study period? Add a description of how to recruit these participants. 

- Please describe in detail the contents of each session of ACT-DEA program.

- Correct the notation as men/women for gender in the results.

Author Response

We appreciate the time and effort the reviewers have dedicated to providing valuable feedback on the manuscript. Please see below, in blue, for a point-by-point response to reviewers' comments and concerns.

Point 1: When is the study period? Add a description of how to recruit these participants.

Response 1: As suggested by the reviewer, we changed “We conducted a quasi-experimental study as a feasibility study for a future randomized controlled trial.” to “We conducted a quasi-experimental study, from September 2019 to February 2020, as a feasibility study for a future randomized controlled trial.” to the Participants part of the Materials and Methods section. We also changed “Participants were in their middle and late adolescence (aged 15-25 years) in South Korea who had shown DEA behavior for the previous three months.” to “Participants were in their middle and late adolescence (aged 15-25 years) who had shown DEA behavior for the previous three months and were recruited from several psychiatric clinics in South Korea.” to the Participants part of the Materials and Methods section.

Point 2: Please describe in detail the contents of each session of ACT-DEA program.

Response 2: We think this is an excellent suggestion. We added, “The details of the program were as follows: (1) Introduction to the ACT matrix and crisis survival skills training (2) Understanding functional analysis of behaviors on the ACT matrix and learning mindful grounding techniques (3) Value clarifications and reinforcing stabilization techniques (4) Identifying the effectiveness of DEA behaviors (5) Understanding acceptance using metaphors (6) Planning commitment behaviors and closing.” to the ACT-DEA program part of the Materials and Methods section.

Point 3: Correct the notation as men/women for gender in the results.

Response 3: Thank you for pointing this out. We made corrections to Table 1.

Reviewer 2 Report

This is a nicely performed study that investigated efficacy and feasibility of structured ACT program that alleviates DEA behaviors for adolescents with clinically significant levels of mental health issues.  Authors have submitted quite a fascinating manuscript and I sure would like to see it printed soon.

1.       I estimated the revised manuscript as “Minor Revisions” from these reasons.

2.       The number of participants listed in the abstract and results is inconsistent. The authors should check your manuscript carefully again.

3.       The author should add at the introduction what approaches have been taken to treat DEA, their effectiveness and issues.

4.       P4, Line 168 – p5, Line 170: “In other words, many of them engaged in their DEA to control their private events such as negative ideas or feelings.” This sentence should be mentioned in the discussion.

5.       Please briefly mention the reasons for the five participants who interruption.

6.       P10, Line 229 ACD-DEA⇒ACT-DEA

Author Response

We appreciate the time and effort the reviewers have dedicated to providing valuable feedback on the manuscript. Please see below, in blue, for a point-by-point response to reviewers' comments and concerns.

Point 1: The number of participants listed in the abstract and results is inconsistent. The authors should check your manuscript carefully again.

Response 1: Thank you for pointing this out. We made a correction to the number of participants listed in the abstract.

Point 2: The author should add at the introduction what approaches have been taken to treat DEA, their effectiveness and issues.

Response 2: We added the suggested content to the Introduction section with a reference: “In this context, it has been required to develop therapeutic interventions for DEA behaviors in adolescents. For instance, a meta-analysis reported the therapeutic possibility of various psychological interventions, including CBT (Cognitive Behavioral Therapy), DBT (Dialectical Behavior Therapy), and MBT (Mentalization-based Therapy), in reducing non-suicidal self-injury in adolescents. However, there remains an urgent need to identify specific and transdiagnostic approaches to overall DEA behaviors.”

Point 3: P4, Line 168 – p5, Line 170: “In other words, many of them engaged in their DEA to control their private events such as negative ideas or feelings.” This sentence should be mentioned in the discussion.

Response 3: We think this is an excellent suggestion. We added, “Those specific therapeutic processes of the ACT-DEA program may improve the intrapersonal discomforts most participants reported as the reason for engaging in their DEA.” to the Discussion section.

Point 4: Please briefly mention the reasons for the five participants who interruption.

Response 4: As suggested by the reviewer, we added “three dropped out due to acute psychiatric hospitalization, and the other two for undisclosed reasons.” to the Characteristics of destructive experiential avoidance part of the Reults section.

Point 5: P10, Line 229 ACD-DEA⇒ACT-DEA?

Response 5: Thank you for pointing this out. We corrected the wording accordingly.

Reviewer 3 Report

In this manuscript the authors investigated the feasibility and efficacy of Acceptance and Commitment Therapy (ACT) for Destructive Experiential Avoidance (DEA) in adolescents with mental health issues.
The topic is certainly interesting and has a major public health echo, that could fit the journal’s area of interest.
The design of the study and the methods are expressed clearly and answer to its aim. The discussion is supported by the results and the interpretation of the data leads to interesting conclusions.
The authors describe very well the impact of DEA on adolescents and young adults life-course, but I suggest for both a clinical and public health oriented perspective they may add the impact that DEA might have on the health systems, in the light of the increasing need of hospitalization for adolescents and young adults. Here below two suggested papers to support this section
1)Bartoli F, Cavaleri D, Moretti F, et al. Pre-Discharge Predictors of 1-Year Rehospitalization in Adolescents and Young Adults with Severe Mental Disorders: A Retrospective Cohort Study. Medicina (Kaunas). 2020;56(11):613. Published 2020 Nov 15. doi:10.3390/medicina56110613
2)Lunde KB, Mehlum L, Melle I, Qin P. Psychiatric admissions after hospital presented deliberate self-harm in the young: A national study. J Psychiatr Res. 2022;151:575-582. doi:10.1016/j.jpsychires.2022.05.020
The authors stated very well the main limitations of this study, i.e. the single arm nature, the lack of long term follow up and the presence of several possible inferences from other variables, for instance pharmacological treatment. I suggest the authors add to this section the non negligeable rate of drop outs (25%).

Author Response

We appreciate the time and effort the reviewers have dedicated to providing valuable feedback on the manuscript. Please see below, in blue, for a point-by-point response to reviewers' comments and concerns.

Point 1: The authors describe very well the impact of DEA on adolescents and young adults life-course, but I suggest for both a clinical and public health oriented perspective they may add the impact that DEA might have on the health systems, in the light of the increasing need of hospitalization for adolescents and young adults. Here below two suggested papers to support this section:

1) Bartoli F, Cavaleri D, Moretti F, et al. Pre-Discharge Predictors of 1-Year Rehospitalization in Adolescents and Young Adults with Severe Mental Disorders: A Retrospective Cohort Study. Medicina (Kaunas). 2020;56(11):613. Published 2020 Nov 15. doi:10.3390/medicina56110613

2) Lunde KB, Mehlum L, Melle I, Qin P. Psychiatric admissions after hospital presented deliberate self-harm in the young: A national study. J Psychiatr Res. 2022;151:575-582. doi:10.1016/j.jpsychires.2022.05.020.

Response 1: We think this is an excellent suggestion. We added, “As previous studies indicated, DEA behaviors in adolescents also increase the public health burden, such as rehospitalizations and emergency department visits.” to the Introduction section. We also appended the papers mentioned above to the References.

Point 2: The authors stated very well the main limitations of this study, i.e. the single arm nature, the lack of long term follow up and the presence of several possible inferences from other variables, for instance pharmacological treatment. I suggest the authors add to this section the non negligeable rate of drop outs (25%).

Response 2: We agree that this is a potential limitation of the study. We have added this as a limitation: “Third, the dropout rate of the study might affect the outcome as well.”